

# From grit to flourishing: physical literacy's mediating role in enhancing well-being among college students with obesity

Xingyu Liu[1], Zidong Li[1], Wanru Cheng[2], Jian Zhang[2], Xiaoyu Ma[2], Di Tang[3], Jinde Liu[4], Tianyu Gao[2], Ting Liu[5], Tao Chen[1] and Ruisi Ma[2]

[1] Badminton Technical and Tactical Analysis and Diagnostic Laboratory, National Academy of Badminton, Guangzhou Sport University, Guangzhou, China
[2] School of Physical Education, Jinan University, Guangzhou, China
[3] The Nethersole School of Nursing, Faculty of Medicine, Chinese University of Hong Kong, Shatin, New Territories, Hong Kong
[4] Faculty of Physical Education, Fudan University, Shanghai, China
[5] School of Nursing, Sun Yat-sen University, Guangzhou, China

Corresponding authors
Tao Chen, 11004@gzsport.edu.cn
Ruisi Ma, penny@link.cuhk.edu.hk

## ABSTRACT

**Objective**. To investigate whether physical literacy mediates the relationship between grit and well-being among college students with obesity.

**Methods**. A total of 385 students with obesity were recruited. Participants completed validated questionnaires measuring grit, physical literacy, and well-being. Mediation analyses were performed to estimate indirect effects and generate bias-corrected 95% confidence intervals (CI).

**Results**. Grit was positively associated with physical literacy, and physical literacy was positively associated with well-being. Physical literacy partially mediated the relationship between grit and well-being, with the indirect effect accounting for 26.32% of the total effect (indirect effect = 0.20, 95% CI [0.09–0.31]). In a parallel mediation model analyzing the subdimensions of physical literacy, the "interaction with the environment" emerged as the strongest mediator (indirect effect = 0.15, 95% CI [0.10–0.21]), accounting for 19.74% of the total effect. The indirect effects through "motivation" and "confidence and physical competence" were also significant but accounted for smaller proportions of the total effect (6.58% and 5.26%, respectively).

**Conclusions**. These findings serve as an initial step in understanding how physical literacy, particularly the ability to interact with the environment, partially mediates the relationship between grit and well-being among college students with obesity. Future interventional research aiming to enhance physical literacy—especially environmental engagement—is needed to confirm whether it can amplify the positive impact of grit on well-being. A multifaceted approach that fosters both psychological traits and physical competencies may prove beneficial in improving the psychological and physical health of this population.

## INTRODUCTION

Obesity among college students has become a significant global public health concern, with prevalence rates increasing alarmingly over the past few decades (*Ng et al., 2014*; *Ward et al., 2019*). This trend is particularly troubling given the numerous adverse health outcomes associated with obesity, including cardiovascular disease, type 2 diabetes, and mental health disorders (*Okunogbe et al., 2021*). College students represent a critical demographic, as habits formed during this transitional period often persist into adulthood, influencing long-term health trajectories (*Nelson et al., 2008*; *KavehFarsani, Kelishadi & Beshlideh, 2020*; *Sun et al., 2021*). Despite the recognition of obesity's physical health implications, there is a pressing need to understand the psychological factors that can enhance well-being in college students with obesity, a population that often experiences stigma, social isolation, and reduced quality of life (*Puhl & Heuer, 2010*; *Smith, Fu & Kobayashi, 2020*; *Pearl, Wadden & Jakicic, 2021*). College students with obesity (body mass index (BMI) $\geq$ 30 kg/m$^2$) represent a particularly vulnerable subgroup, as they not only face elevated risks of chronic diseases but also more pronounced weight-related stigma, which can negatively influence mental health, self-esteem, and social engagement (*Yu-xiang, Hui & Chong-lin, 2023*; *Srishti & Pareek, 2024*). Existing literature suggests that obesity in emerging adulthood may exacerbate these psychological stressors (*Dąbrowska et al., 2020*), underscoring the need to examine specific protective factors—such as grit and physical literacy—that could help buffer negative outcomes and improve well-being for this population (*She et al., 2023*).

Grit, defined as perseverance and passion for long-term goals, has been identified as a crucial trait that contributes to success and well-being across various populations (*Duckworth & Quinn, 2009*). Empirical studies have linked higher levels of grit to enhanced academic achievement, lower attrition rates, and improved psychological well-being in higher education settings (*Bowman et al., 2015*; *Credé, Tynan & Harms, 2017*). Individuals exhibiting high levels of grit tend to maintain motivation and resilience, contributing positively to their overall life satisfaction and mental health (*Datu, Valdez & King, 2016*; *Disabato, Goodman & Kashdan, 2019*). For students with obesity, grit may play an essential role in navigating the additional hurdles they may encounter, potentially buffering against negative outcomes and fostering psychological flourishing (*Totosy De Zepetnek et al., 2021*)

Well-being is a multidimensional construct that reflects optimal psychological functioning and experience (*Ryan & Deci, 2001*). The PERMA model, proposed by Seligman, conceptualizes well-being through five core elements: Positive Emotion, Engagement, Relationships, Meaning, and Accomplishment (*Seligman, 2011*). This model provides a comprehensive framework for understanding how individuals thrive and flourish. In the context of college students, well-being, as defined by the PERMA model, is particularly relevant, as it encapsulates emotional well-being, involvement in meaningful activities, the quality of social relationships, a sense of purpose, and the achievement of personal goals (*Butler & Kern, 2016*). Prior research has indicated that higher levels of grit are associated with enhanced well-being across these dimensions, suggesting that gritty individuals may experience greater life satisfaction and psychological flourishing

(*Vainio & Daukantaite, 2016*). Obese individuals may experience disparities in these well-being dimensions due to societal pressures and health-related challenges (*Lin et al., 2020*). Investigating factors that enhance well-being in this group is therefore of critical importance.

Simultaneously, physical literacy has emerged as a holistic construct encompassing the motivation, confidence, physical competence, knowledge, and understanding necessary to value and engage in physical activities throughout life (*Whitehead, 2010*; *Edwards et al., 2017*). Physical literacy extends beyond physical capabilities to include cognitive and affective dimensions that foster lifelong participation in physical activity, which is crucial for both physical and mental well-being (*Robinson et al., 2015*). Engaging in regular physical activity has been linked to improved well-being as per the PERMA model, contributing to positive emotions, opportunities for engagement, enhanced relationships through social interactions, a sense of meaning, and feelings of accomplishment (*Lubans et al., 2016*). For college students with obesity, physical literacy may be particularly pertinent, as it can influence their willingness and ability to participate in physical activity, which is crucial for both physical and mental health (*Cairney et al., 2019*; *Ma et al., 2020b*; *Ma et al., 2021*).

From a cultural perspective, collectivist societies such as China place strong emphasis on social harmony, family expectations, and academic achievement (*Zhang & Han, 2023*). These cultural factors can shape the development of grit and physical literacy in unique ways (*Li, Fan & Leong, 2023*). For example, students may feel compelled to persist in group-based physical activities due to social norms or familial encouragement (*Bao et al., 2020*). Likewise, the sense of "interaction with the environment" in physical literacy may be influenced by communal exercise settings, peer interactions, or campus cultures that promote unity and collective goals. Understanding how these cultural nuances interact with individual traits like grit can offer deeper insight into strategies to enhance well-being in Chinese college students with obesity (*Xie, Fan & He, 2023*).

The potential mediating role of physical literacy in the relationship between grit and well-being among college students with obesity presents a promising yet underexplored area of inquiry. It is posited that individuals with higher levels of grit are more likely to engage persistently in physical activities, thereby developing greater physical literacy, which in turn enhances their well-being across the PERMA dimensions (*Tedesqui & Young, 2017*). Gritty college students with obesity may exhibit a propensity for sustained engagement in physical activities despite challenges, leading to the development of physical competence and confidence—key components of physical literacy (*Howard & Crayne, 2019*; *Guerrero et al., 2019*). This sustained engagement not only reinforces physical literacy but also contributes to improved well-being through mechanisms such as enhanced self-efficacy, stress reduction, social connectedness, and a sense of achievement (*Wilson, Ellison & Cable, 2015*; *Lubans et al., 2016*).

Understanding this mediation is crucial, as it could inform targeted interventions aimed at promoting both psychological and physical health among college students with obesity. By fostering physical literacy, educational institutions can enhance the well-being of students with high levels of grit, enabling them to channel their perseverance into beneficial activities that support weight management and psychological flourishing (*Jefferies et al.,*

**Table 1  Demographic and physical characteristics of the participants.**

| Characteristics | Male ($n = 233$), Mean (SD) | Female ($n = 152$), Mean (SD) | Total ($N = 385$), Mean (SD) |
|---|---|---|---|
| Age (Years) | 19.50 (0.82) | 19.60 (0.86) | 19.54 (0.84) |
| BMI (kg/m$^2$) | 31.26 (1.01) | 31.04 (1.09) | 31.17 (1.04) |
| GPA (5.0) | 2.93 (0.50) | 2.92 (0.47) | 2.93 (0.48) |
| Grade (n, %) | | | |
|     Grade 1 | 122 (52.36%) | 75 (49.34%) | 197 (51.17%) |
|     Grade 2 | 71 (30.47%) | 50 (32.89%) | 121 (31.43%) |
|     Grade 3 | 38 (16.31%) | 25 (16.45%) | 63 (16.36%) |
|     Grade 4 | 2 (0.86%) | 2 (1.32%) | 4 (1.04%) |
| Only child in family (n, %) | | | |
|     Yes | 222 (95.28%) | 146 (96.05%) | 368 (95.58%) |
|     No | 11 (4.72%) | 6 (3.96%) | 17 (4.42%) |

Notes.
BMI, Body Mass Index; GPA, Grade Point Average.

*2019*; *Jeong & So, 2020*). Programs that integrate physical literacy development with support for building grit may offer a multifaceted approach to improving health outcomes in this population (*Schure, Christopher & Christopher, 2008*; *Caldwell et al., 2009*).

Therefore, this study aims to investigate the mediating role of physical literacy in the relationship between grit and well-being among college students with obesity. We hypothesize that physical literacy serves as a potentially important mediator, whereby college students with obesity with higher levels of grit are more likely to develop physical literacy, which in turn enhances their overall well-being.

## METHODS

### Participants

A total of 385 obese undergraduate students were recruited from six major universities in Guangzhou, China: Sun Yat-sen University, South China Normal University, Jinan University, South China University of Technology, Guangzhou University, and South China Agricultural University. Participants ranged in age from 18 to 22 years ($M = 19.54$, standard deviation (SD) $= 0.84$), with 60.52% identifying as male and 39.48% as female. Inclusion criteria required participants to be enrolled as full-time undergraduate students and classified as obese, defined by a body mass index (BMI) $\geq 30$ kg/m$^2$ according to the World Health Organization (WHO) (*World Health Organization, 2024*). Table 1 presents the detailed characteristics of the participants.

### Recruitment and sampling

Participants were recruited over a six-month period *via* university health centers, student associations, and campus-wide email announcements. We first obtained a list of students identified as having obesity during routine health check-ups or through self-report. Invitations were then randomly distributed *via* emails and posters, ensuring a balanced approach across the six universities. Although the final sample had a somewhat higher

proportion of males (60.52%), this distribution aligns with the response rates observed in our target population within these universities. Students were eligible if they were (a) full-time undergraduates and (b) classified as having obesity (BMI $\geq$ 30 kg/m$^2$). Written informed consent was obtained from all participants.

## Measures

### Grit

Grit was assessed using the Short Grit Scale (Grit-S) developed by Duckworth (*Duckworth & Quinn, 2009*). The Chinese version of the Grit-S, which has been translated and validated for reliability and validity among Chinese populations (*Li et al., 2018*; *Zhong et al., 2018*), was employed in this study. The Grit-S is an 8-item self-report instrument measuring two dimensions: Perseverance of Effort (*e.g.*, "I am a hard worker") and Consistency of Interests (*e.g.*, "My interests change from year to year" [reverse-scored]). Participants respond on a 5-point Likert scale ranging from 1 (not at all like me) to 5 (very much like me). The Grit-S has demonstrated good internal consistency (Cronbach's $\alpha = 0.86$) and construct validity (0.73 to 0.83) (*Duckworth & Quinn, 2009*). In the present study, the overall Cronbach's alpha was 0.81.

### Physical literacy

Physical literacy was assessed using the Simplified Chinese Version of the Perceived Physical Literacy Instrument (PPLI-SC) (*Ma et al., 2020a*), which encompasses three dimensions: (1) Motivation (*e.g.*, "I aspire to know the current sports trend"), (2) Confidence and Physical Competence (*e.g.*, "I possess adequate fundamental movement skills"), and (3) Interaction with the Environment (*e.g.*, "I have strong social skills"). Developed based on Whitehead's definition of physical literacy, the instrument measures college students' attitudes toward physical activity and the extent to which they take responsibility for their bodies (*Whitehead, 2010*). Notably, the "Interaction with the Environment" subdimension captures participants' subjective perception of their ability to engage effectively in social and physical surroundings, rather than objectively assessing actual environmental interaction. For instance, an item like "I have strong social skills" reflects how respondents perceive their capacity for interpersonal engagement in physical activity contexts, rather than any standardized measure of social competence. The Simplified Chinese Version has demonstrated strong reliability and validity for measuring physical literacy among Chinese university students, with a Cronbach's $\alpha$ of 0.86 and confirmatory factor analysis indicating factor loadings ranging from 0.60 to 0.92 (*Ma et al., 2020b*). In this study, the overall Cronbach's alpha was 0.84.

### Well-being

Well-being was evaluated using the PERMA-Profiler (*Butler & Kern, 2016*), a 23-item self-report measure assessing the five dimensions of the PERMA model: Positive Emotion, Engagement, Relationships, Meaning, and Accomplishment. The translated Chinese version of the PERMA-Profiler demonstrated excellent cultural adaptability. It exhibited high Cronbach's alpha coefficients ($\alpha = 0.79$–$0.88$), indicating strong internal consistency, as well as good divergent validity ($r = -0.19$ to $-0.38$) and convergent validity ($r =$

0.53–0.85). Structural validity was also satisfactory (*Nie et al., 2024*). In the current study, the overall Cronbach's alpha was 0.89.

## Procedure

The study was conducted in accordance with the ethical standards of the Declaration of Helsinki and was approved by the Institutional Review Board of Jinan University (Approval Number: JNUKY-2023-0154). According to the approved protocol, this cross-sectional survey study was part of a broader project investigating exercise training and physical literacy in college students. Ethical approval covered both the intervention and any secondary analyses arising from the data. Interested students completed a screening session where trained research assistants measured height and weight to confirm BMI. After eligibility was confirmed, participants were provided with a secure link to an online survey that included demographic items and the main instruments (Grit-S, PPLI-SC, and the PERMA-Profiler). The presentation of the main instruments was randomized to minimize order effects, and all surveys were completed anonymously online.

Although random invitations were distributed across all six universities, a notably higher proportion of male students ($n = 233$) than female students ($n = 152$) ultimately enrolled in the study. This imbalance appears to reflect underlying demographic patterns observed in the participating institutions: health screening records showed a slightly higher prevalence of obesity among male undergraduates, and a greater willingness among male students to volunteer for weight-related health research (*Karnes et al., 2021*). Because no stratification by sex was imposed during recruitment, these factors led to a naturally higher male-to-female ratio in our sample.

## Sample size

The study size was determined based on a priori power analysis to detect a medium effect size ($f^2 = 0.15$) in the mediation model with three predictors, using G*Power 3.1 software. A minimum of 143 participants was required to achieve 80% power at a 0.05 significance level. To account for potential dropouts, incomplete data , and to allow for further subgroup analyses, the target sample size was increased. Ultimately, 385 participants were successfully recruited, providing even greater power and robustness for our mediation analyses.

## Data analysis

All statistical analyses were performed using the Statistical Package for the Social Sciences (SPSS, version 26.0). Descriptive statistics (means, standard deviations) were calculated for grit, physical literacy, and well-being (PERMA) scores. Skewness and kurtosis values were examined to assess the normality of data distributions. Bivariate Pearson correlation analyses were conducted to explore the relationships among grit, well-being, and physical literacy dimensions (Motivation, Confidence and Physical Competence, and Interaction with the Environment). These correlations provided preliminary insights into the associations between the variables. To test the mediating effect of physical literacy on the relationship between grit and well-being, the SPSS PROCESS macro (version 3.5) was employed. Specifically, Model 4 of the PROCESS macro was utilized to assess the mediation effect. Bootstrapping procedures with 5,000 resamples were applied to generate

**Table 2   Descriptive statistics and bivariate correlations.**

|  | Mean (SD) | 1 | 2 | 3 | 4 | 5 | 6 |
|---|---|---|---|---|---|---|---|
| 1. Grit | 3.21 (0.32) | – |  |  |  |  |  |
| 2. Well-Being (PERMA) | 156.57 (24.36) | 0.25* | – |  |  |  |  |
| 3. Motivation | 10.77 (2.19) | 0.42* | 0.45* | – |  |  |  |
| 4. Confidence and Physical Competence | 10.67 (2.28) | 0.48* | 0.45* | 0.79* | – |  |  |
| 5. Interaction with the Environment | 7.65 (1.32) | 0.44* | 0.40* | 0.60* | 0.64* | – |  |
| 6. Physical literacy | 29.10 (5.19) | 0.49* | 0.48* | – | – | – | – |

**Notes.**

Correlation coefficients are presented below the diagonal.

*Indicates $p < 0.001$.

bias-corrected 95% confidence intervals (CIs) for the indirect effects. An indirect effect was considered statistically significant if the 95% CI did not include zero. This non-parametric resampling method enhances the robustness of the mediation analysis, particularly in cases where normality assumptions may not hold. All statistical tests were two-tailed, with a significance level set at $p < 0.05$.

## RESULTS

### Preliminary analyses

All participants provided complete and valid responses for all measures, with no missing data detected. Descriptive statistics and correlations among grit, well-being, and physical literacy (including the subdimensions) are presented in Table 2. The skewness (|skewness| < 2) and kurtosis (|kurtosis| < 7) values indicated that the data were approximately normally distributed. To control for Type I error due to multiple comparisons, we adjusted the significance threshold using the Bonferroni correction, setting it at $0.05/6 = 0.008$. Because the physical literacy score is the sum of its three subdimensions, we did not perform correlation analyses between each subdimension and physical literacy itself, in order to avoid redundancy and potential issues with multicollinearity.

### Mediation analysis using total physical literacy score

The PROCESS macro (Model 4) was employed to test the mediation effect of physical literacy between grit and well-being among college students with obesity. Grit was designated as the focal predictor (X); physical literacy was the proposed mediator (M). X and M were incorporated as predictors in an ordinary least-squares regression analysis to predict the students' well-being (Y). Building upon the following regression equations, the results of the bootstrapping analysis are presented in Table 3.

$$Y = a_1 + cX + \varepsilon_1 \tag{1}$$

$$M = a_2 + aX + \varepsilon_2 \tag{2}$$

$$Y = a_3 + c'X + bM + \varepsilon_3 \tag{3}$$

where a, b, and c' estimate the paths separately.

**Table 3  Mediating effects of grit on well-being through physical literacy.**

|  | Effect value | SE | LLCI | ULCI | Effect size |
|---|---|---|---|---|---|
| Total effect | 0.76 | 0.07 | 0.63 | 0.89 | |
| Direct effect | 0.56 | 0.06 | 0.45 | 0.67 | 73.68% |
| Indirect effect | 0.20 | 0.06 | 0.09 | 0.31 | 26.32% |

**Notes.**

SE, Standard Error; LLCI, Lower Limit of the Confidence Interval; ULCI, Upper Limit of the Confidence Interval.

The mediation analysis demonstrated that grit was significantly positively associated with physical literacy ($\beta = 0.12$, $F = 28.81$, $p < 0.001$). Physical literacy was also significantly positively associated with well-being ($\beta = 0.56$, $F = 925.94$, $p < 0.001$). The total effect of grit on well-being was significant (total effect = 0.76, 95% CI [0.63–0.89]), and when physical literacy was included in the model, the direct effect of grit on well-being remained significant but was reduced (direct effect = 0.56, 95% CI [0.45–0.67]), accounting for 73.68% of the total effect. The indirect effect of grit on well-being through physical literacy was significant (indirect effect = 0.20, 95% CI [0.09–0.31]), accounting for 26.32% of the total effect; since the bootstrap confidence interval did not include zero, this confirms that physical literacy partially mediates the relationship between grit and well-being among college students with obesity.

## Parallel mediation analysis using physical literacy subdimensions

To further investigate the mediating effects of the subdimensions of physical literacy on grit and well-being, we employed a parallel mediation model. This analytical approach tested the mediating effects of the three subdimensions of physical literacy—motivation ($M_1$), confidence and physical competence ($M_2$), and interaction with the environment ($M_3$)—on the relationship between grit (X) and well-being (Y).

$$M_1 = a_1 X + \varepsilon_1 \tag{4}$$

$$M_2 = a_2 X + \varepsilon_2 \tag{5}$$

$$M_3 = a_3 X + \varepsilon_3 \tag{6}$$

$$Y = c' X + b_1 M_1 + b_2 M_2 + 3 M_3 + \varepsilon_4, \tag{7}$$

where $a_1$, $a_2$, $a_3$ are the coefficients representing the effect of grit on each mediator. $b_1$, $b_2$, $b_3$ are the coefficients representing the effect of each mediator on well-being. $c'$ is the direct effect of grit on well-being after accounting for the mediators.

The results of the parallel mediation analysis are presented in Tables 4 and 5. Figure 1 presents the graphical representation of the mediation model and the regression coefficients. As shown in Table 4, grit was significantly positively associated with all three subdimensions of physical literacy: motivation ($\beta = 0.07$, $F = 10.86$, $p < 0.001$), confidence and physical competence ($\beta = 0.08$, $F = 12.20$, $p < 0.001$), and interaction with the environment

**Table 4** Regression analyses in the parallel mediation model.

| Independent variable | Dependent variable | β | F | R² | ΔR² |
|---|---|---|---|---|---|
| Grit | Motivation | 0.07* | 10.86 | 0.03 | 0.01 |
| | Confidence and physical competence | 0.08* | 12.20 | 0.04 | 0.02 |
| | Interaction with the environment | 0.21* | 93.17 | 0.11 | 0.06 |
| | Well-Being (PERMA) | 0.24* | 123.27 | 0.06 | 0.03 |
| Motivation | Well-Being (PERMA) | 0.50* | 533.21 | 0.20 | 0.18 |
| Confidence and physical competence | Well-Being (PERMA) | 0.49* | 652.847 | 0.24 | 0.23 |
| Interaction with the environment | Well-Being (PERMA) | 0.50* | 672.04 | 0.25 | 0.24 |

Notes.
*Indicates $p < 0.001$.

**Table 5** Mediating effects of grit on well-being through physical literacy subdimensions.

| | Effect value | SE | LLCI | ULCI | Effect size |
|---|---|---|---|---|---|
| Total effect | 0.76 | 0.07 | 0.63 | 0.89 | – |
| Direct effect | 0.51 | 0.06 | 0.40 | 0.63 | 67.11% |
| M1 | 0.05 | 0.02 | 0.01 | 0.09 | 6.58% |
| M2 | 0.04 | 0.02 | 0.01 | 0.08 | 5.26% |
| M3 | 0.15 | 0.03 | 0.10 | 0.21 | 19.74% |
| C1 | −0.01 | 0.02 | −0.04 | 0.03 | – |
| C2 | −0.11 | 0.02 | −0.15 | −0.07 | – |
| C3 | −0.10 | 0.02 | −0.15 | −0.06 | – |

Notes.
M1, Indirect effect of motivation; M2, Indirect effect of confidence and physical competence; M3, Indirect effect of interaction with the environment; C1, M2–M1; C2, M2–M3; C3, M1–M3.

($\beta = 0.21$, $F = 93.17$, $p < 0.001$). Each subdimension was also significantly positively associated with well-being. Table 5 shows the mediating effects; the total effect of grit on well-being was significant (effect value = 0.76, standard error (SE) = 0.07, lower limit of the confidence interval (LLCI) = 0.63, upper limit of the confidence interval (ULCI) = 0.89), and when considering the mediators, the direct effect of grit on well-being remained significant but was reduced (effect value = 0.51, LLCI = 0.40, ULCI = 0.63), accounting for 67.11% of the total effect. The indirect effects through the subdimensions of physical literacy were all significant: motivation (indirect effect = 0.05, 95% CI [0.01–0.09]), accounting for 6.58% of the total effect; confidence and physical competence (indirect effect = 0.04, 95% CI [0.01–0.08]), accounting for 5.26% of the total effect; and interaction with the environment (indirect effect = 0.15, 95% CI [0.10–0.21]), accounting for 19.74% of the total effect.

## DISCUSSION

This study aimed to investigate the mediating role of physical literacy in the relationship between grit and well-being among college students with obesity. The findings provide preliminary evidence that physical literacy partially mediates this relationship, highlighting the importance of physical literacy in enhancing the well-being of this population. Notably,

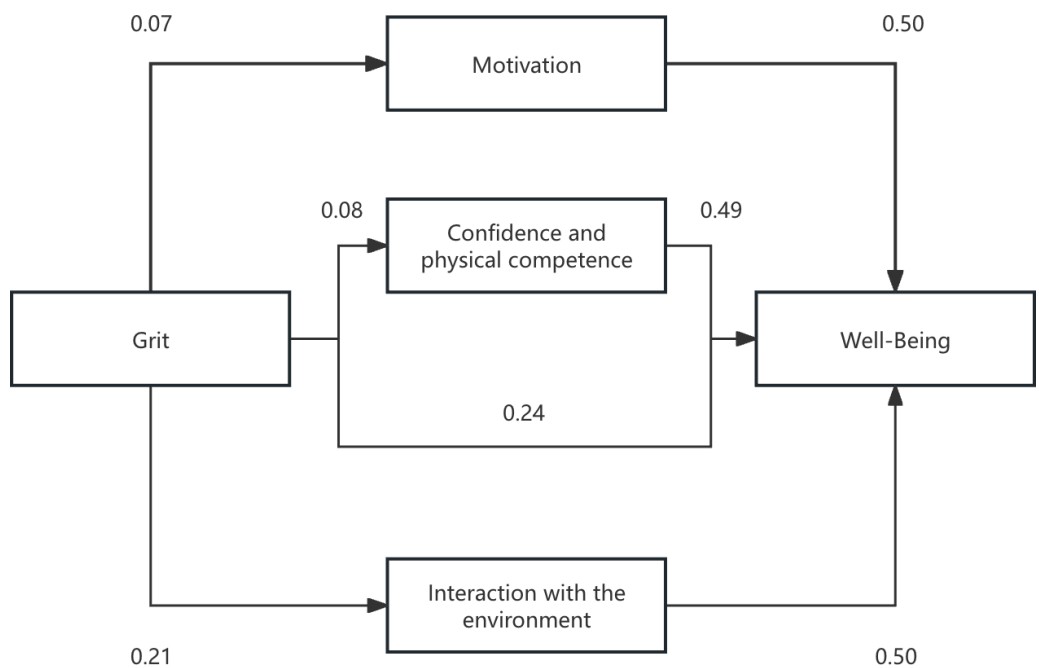

**Figure 1** **Graphical illustration of the parallel mediation model.**

among the subdimensions of physical literacy, interaction with the environment emerged as the most significant mediator, suggesting that engagement with one's physical and social environment is crucial in translating grit into well-being.

Consistent with our hypotheses and previous research, we found that grit was significantly positively associated with well-being among college students with obesity. This aligns with studies indicating that individuals with higher levels of grit tend to experience greater psychological well-being, life satisfaction, and resilience to adversity (*Duckworth & Quinn, 2009*; *Vainio & Daukantaite, 2016*; *Disabato, Goodman & Kashdan, 2019*). Grit, in many respects, underpins the development of resilience, which involves adapting positively to challenges and setbacks (*Jefferies, 2022*). Gritty individuals are more likely to persevere through challenges and maintain consistent interests over time, facilitating a form of "positive challenge" engagement may buffer against the psychological stressors associated with obesity, such as stigma, discrimination, and social isolation (*Puhl & Heuer, 2010*; *Totosy De Zepetnek et al., 2021*).

The mediation analysis extends the existing literature by demonstrating that physical literacy is a significant pathway through which grit enhances well-being. Specifically, higher levels of grit were associated with greater physical literacy, which in turn was linked to improved well-being. This finding supports the notion that physical literacy is not merely about physical competence but also encompasses motivation, confidence, and the ability to interact effectively with the environment (*Whitehead, 2010*; *Edwards et al., 2017*). Gritty individuals may be more motivated to engage in physical activities, persist in developing

physical skills, and overcome barriers to participation, thereby enhancing their physical literacy (*Tedesqui & Young, 2017*; *Guerrero et al., 2019*).

Among the subdimensions of physical literacy, interaction with the environment had the strongest mediating effect. This suggests that the ability to engage with the physical and social environment is particularly important for the well-being of college students with obesity. Engaging in physical activities often involves social interaction, teamwork, and a sense of belonging, which are key components of the PERMA model of well-being (*Seligman, 2011*; *Butler & Kern, 2016*). Participation in group sports or recreational activities can foster positive emotions, build meaningful relationships, provide a sense of accomplishment, and enhance engagement—all contributing to overall well-being (*Lubans et al., 2016*; *Cairney et al., 2019*).

In Chinese culture, collectivist norms often emphasize social harmony, group cohesion, and interdependence (*Liu et al., 2021*). Students may feel motivated to persist in physical activities that involve peer collaboration or institutional support (*Sheng, Gong & Zhou, 2023*). This cultural backdrop may also reinforce the importance of "interaction with the environment" within physical literacy. For instance, campus-wide fitness initiatives, group sports, or university-sponsored health campaigns can provide avenues for students to demonstrate their grit in a socially accepted framework. Such participation not only helps mitigate the adverse effects of weight stigma but also fosters communal relationships, a sense of accomplishment, and meaningful engagement—all of which are central pillars of the PERMA model (*Seligman, 2011*; *Oraibi et al., 2024*).

Beyond these cultural and interpersonal considerations, negative psychological traits such as anxiety or depressive symptoms may also shape how college students with obesity engage in physical literacy and derive benefits from grit. Recent studies emphasize the role of discrete emotions in predicting cognitive and physical achievements (*Simonton, Wallhead & Kern, 2024*), suggesting that students who experience more negative affect may struggle to maintain motivation, self-esteem, and consistent participation in physical activities (*Simonton & Layne, 2023*; *Woolley, Houser & Kriellaars, 2024*). Conversely, positive emotional states can reinforce enjoyment and bolster the perceived value of physical literacy, creating a virtuous cycle in which students persist in health-promoting behaviors. These findings underscore the importance of addressing emotional well-being alongside physical literacy and grit, as targeted interventions that help students regulate negative emotions could further enhance the protective benefits of perseverance and environmental engagement.

The significant mediating effects of motivation and confidence and physical competence, although smaller, also underscore their roles in the grit–well-being relationship. Motivation drives individuals to initiate and sustain physical activity, while confidence and physical competence enable them to participate effectively and enjoy the benefits of such activities (*Robinson et al., 2015*; *Ma et al., 2020b*). These components may help college students with obesity overcome psychological barriers, such as fear of judgment or failure, enhancing their willingness to engage in health-promoting behaviors (*Howard & Crayne, 2019*; *Ma et al., 2021*).

## Limitations and future directions

Despite the valuable insights, several limitations should be acknowledged. The cross-sectional design limits the ability to infer causality; longitudinal studies are needed to establish temporal relationships and examine how changes in grit and physical literacy over time impact well-being. Additionally, the reliance on self-report measures may introduce social desirability and recall biases; future research could incorporate objective measures of physical activity and physical competence, such as accelerometers or fitness tests, to validate self-reported data. The sample was limited to college students with obesity from universities in Guangzhou, China, which may affect the generalizability of the findings; cultural factors may influence perceptions of grit, physical literacy, and well-being, so replicating the study in different cultural contexts and including diverse populations would enhance the external validity of the results. Furthermore, while the study focused on physical literacy as a mediator, other potential mediators or moderators—such as social support, self-esteem, or mental health literacy—were not examined; exploring these factors could provide a more comprehensive understanding of the mechanisms linking grit and well-being.

## Practical implications

These findings offer an important starting point for colleges and universities seeking to enhance psychological and physical well-being among students with obesity. Interventions that simultaneously address grit and physical literacy can be woven into existing campus frameworks, such as health centers and counseling services that incorporate resilience training, mindfulness, or goal-setting workshops designed to build perseverance; physical education departments that develop inclusive, skill-building courses and group-based sports aimed at reinforcing self-perceived competence and environmental engagement; and student affairs or campus recreation offices that organize peer-led fitness groups and social sport clubs to facilitate consistent participation and mitigate weight-related stigma. Community partnerships further extend resource availability for those who prefer off-campus settings, potentially reducing barriers to exercise and offering diverse opportunities to maintain motivation. By creating supportive social and physical environments, educators and administrators empower students with obesity to harness their grit more effectively, enabling them to persist through challenges while cultivating a stronger sense of accomplishment, connection, and overall well-being (*Jefferies et al., 2019*).

## CONCLUSIONS

This study highlights that physical literacy—especially the self-perceived ability to interact with environment—may act as an important mediator between grit and well-being among college students with obesity. While these results underscore the potential benefits of fostering both psychological traits and physical competencies, the cross-sectional design does not permit causal inferences. Moreover, "interaction with the environment" reflects subjective perception rather than objective measures of physical or social engagement. Future research, particularly longitudinal or intervention-based studies that incorporate more direct assessments of environmental interaction, will help clarify whether enhancing

physical literacy and grit can actively improve well-being in this population. Despite these limitations, the present findings point to the promise of integrative approaches that address both mental fortitude and physical literacy within higher education settings, potentially contributing to a more holistic model of health and resilience for students with obesity.

## ACKNOWLEDGEMENTS

We would like to express our sincere gratitude to all the research assistants who dedicated their time and effort to data collection for this study. Their commitment and diligence were instrumental in the successful completion of this research. Without their invaluable support, this work would not have been possible.

### Funding

The authors received no funding for this work.

### Competing Interests

The authors declare there are no competing interests.

### Author Contributions

- Xingyu Liu conceived and designed the experiments, authored or reviewed drafts of the article, and approved the final draft.
- Zidong Li conceived and designed the experiments, authored or reviewed drafts of the article, and approved the final draft.
- Wanru Cheng performed the experiments, prepared figures and/or tables, and approved the final draft.
- Jian Zhang performed the experiments, prepared figures and/or tables, and approved the final draft.
- Xiaoyu Ma performed the experiments, prepared figures and/or tables, and approved the final draft.
- Di Tang analyzed the data, authored or reviewed drafts of the article, and approved the final draft.
- Jinde Liu analyzed the data, authored or reviewed drafts of the article, and approved the final draft.
- Tianyu Gao conceived and designed the experiments, prepared figures and/or tables, authored or reviewed drafts of the article, and approved the final draft.
- Ting Liu conceived and designed the experiments, authored or reviewed drafts of the article, and approved the final draft.
- Tao Chen conceived and designed the experiments, prepared figures and/or tables, authored or reviewed drafts of the article, and approved the final draft.
- Ruisi Ma conceived and designed the experiments, analyzed the data, authored or reviewed drafts of the article, and approved the final draft.

## Human Ethics

The following information was supplied relating to ethical approvals (i.e., approving body and any reference numbers):

Institutional Review Board of Jinan University

## Ethics

The following information was supplied relating to ethical approvals (i.e., approving body and any reference numbers):

Institutional Review Board of Jinan University approval to carry out the study (Approval Number: JNUKY-2023-0154)

## Data Availability

The data is available at Harvard Dataverse: Ma, Rui Si, 2024, ''Dataset on Grit, Well-Being, and Physical Literacy Among University Students'', https://doi.org/10.7910/DVN/XU3HGV, Harvard Dataverse, V1.

## Supplemental Information

Supplemental information for this article can be found online at http://dx.doi.org/10.7717/peerj.19382#supplemental-information.

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
