# Peer review of "From grit to flourishing: physical literacy’s mediating role in enhancing well-being among college students with obesity"

_PeerJ, doi:10.7717/peerj.19382_

## Round 0.1 · original submission · Major Revisions

Dear Authors,

Thank you for your submission on such an interesting topic. Please address all of the reviewers' comments. Additionally, I suggest incorporating the following key points to further strengthen your study:

-Include a discussion on how Chinese cultural factors may impact the constructs of grit, physical literacy, and well-being.
-Consider the limitations of self-report measures and suggest incorporating objective measures of physical activity, such as accelerometry, or fitness in future studies.

I would also recommend that you use the so called person-first language throughout the paper. Person-first language emphasizes the individual rather than their condition. For example, instead of "obese college students," use "college students with obesity." This approach is more respectful and aligns with current best practices in health communication, as well as contemporary standards in obesity research.

Reviewer 1 ·

Basic reporting

The relationship between PL and grit and well being in college students is interesting, however the rationale for only recruiting students with BMI > 30 is not well articulated in the introduction. This needs to be more clearly justified as to why the entire college population wasn’t studied – I am not convinced about the targeted population by the rationale provided. The average college student adherence to PA guidelines would be very low, independent of obesity status

Clearly multiple authors wrote different sections of the paper without a full coherence check (see your line at 61). A thorough read through is required. Please see some comments below regarding this. There is a fair bit of reaching in certain sections.

The references are good, a few more could be added that would bolster the findings of the study. Also a stronger justification for the target population is needed.

Introduction

Line 61 – remove “add your introduction here”

Line 79 - relationship of grit to resilience should also be stated here, that PL and been tied to resilience and overcoming challenges

Line 135 - remove the word "crucial" as although this is your hope, the experimental design does not allow for the evaluation of "crucial" in this regard

line 136 - persistent engagement in physical activity – this is conjecture at this time. Revise.

Experimental design

Method


It is important to examine the impact of sex on the associations and in the mediation analysis, as many other PL studies and competency studies have shown sex as assigned at birth to be a factor. Sex differences should also be reported in Table 1.

Also, given the randomizing used why was it that there was a large separation in recruitment numbers of males and females. Please explain.

How was randomized sampling achieved across universities? You need to explain this very clearly or revise. Line 206

How was the 50% increase in sample size determined? Please provide the rationale for this, especially since this x-sect study was part of an interventional trial (as stated in ethics).

I would like to see the actual normal stats – Shapiro Wilk for all the measures reported along with the skew and kurtosis.

Using a Bonferroni style is not needed if the actual P values are reported.

I would like to see correlations of the sub-dimensions (line 245) as this is relevant to the study interpretation (you extract single items -like environmental participation perception) and the reader can deal with the issues of multi-collinearity when interpreting.

“ability to interact with the environment” is a perception not the actual and throughout the manuscript this needs to be made clear, it would be handy to state the wording of the question in the methodology as you do for other items.


This manuscript was ethically approved under the title of “The Effect of Exercise Training on the Physical Literacy and Performance” and as such the current title of the submitted manuscript appears to be a secondary analysis which in my jurisdiction requires additional ethical approval.

The data availability statement is not supported by the consent form nor the ethics documents related to confidentiality (esp since it was anonymous). Please provide the dataset for transparency and replication.

Validity of the findings

Line 377 – there is older and recent data showing the relationship of PL to self-esteem and motivation (for instance the play self validation paper by Jeffries in 2021). This also shows the relationship of perceived environmental participation to actual participation which would bolster your findings.

Figure 1 serves very little purpose and can be stated in text in results, and this is not a “model” per se, just a mediation path diagram. Also having it referenced in the introduction is not appropriate (line 137).

Line 357 – the link of grit and resilience should be made more explicitly here. You already cite Jeffries and Jeffries has an excellent paper on how this can be achieved which aligns with your practical implementation statements

Jefferies, Philip. (2020). Physical literacy and resilience: the role of positive challenges [Jefferies, P. (2020). Physical literacy and resilience: the role of positive challenges. Sciences & Bonheur, 5, 11-26.]. 11-26. 10.17605/OSF.IO/SBZYW.

And this paragraph on practical implementations makes more sense after the limitations and future directions actually leading into a soften conclusion (as stated below). Also, since this is a college context how and where would these interventions be implemented – at university and if so, by whom or what department, in the community?

Limitations

Limitations are reasonably well stated.

The point about negative psychological traits could stand to be listed in the discussion as a paragraph rather than in limitations and perhaps link this to recent studies by Simonton or by Woolley (which also relates to self-esteem and PL) on positive and negative discrete emotions.

Simonton, K. L., Wallhead, T., & Kern, B. (In press). Relationships between students’ emotional experiences and cognitive and physical achievement during a middle school hybrid Sport Education Tactical Model season. Journal of Teaching Physical Education.

Woolley A, Houser N, Kriellaars D. Investigating the relationship between emotions and physical literacy in a quality physical education context. Appl Physiol Nutr Metab. 2024 Dec 1;49(12):1658-1665. doi: 10.1139/apnm-2024-0082. Epub 2024 Aug 19. PMID: 39159488.

Simonton, K. L., & Layne, T. (2023). Investigating middle school students physical education emotions, emotional antecedents, self-esteem, and intentions for physical activity. Journal of Teaching in Physical Education, 42, 757-766.




Conclusions

This conclusion over reaches in using cross-sectional associations and mediation analysis in stating PL has a “pivotal role” line 383. Further, the directionality of the association is also not known, and the authors suggest fostering competencies and grit will enhance well being and this is not a interventional trial, so they need to tone this down and suggest that these findings support examining the relationship through intervention. Finally, the perception of interaction with the environment is not the actual interaction with the environment related to movement experiences, and this needs to be factored in in the conclusion.

Simply put these associations and mediations are important starting points for design of interventional studies as stated in line 124. Mediation and association are the first steps … Indeed the conclusion of the abstract is more fitting with the findings that the over-reaching conclusion of the paper. This difference also reveal an incoherency in the paper, as if multiple authors wrote sections independently and never did a coherency read through (i.e. Line 61 statement).

·

Basic reporting

• Language and Clarity: The manuscript is well-written in professional and clear English. The structure adheres to PeerJ's standards, with a logical flow from the abstract to the conclusion. However, there are some minor refinements needed in the manuscript. For example: "over the past few 63 decades(Ward et al. 2019; Ng et al. 2014)" and “Table 2 1. Grit 3.21(0.32)” lack a space before the word ‘(’.

• Context and Background: n.p.

• Figures and Tables: n.p.

• Raw Data: No.
According to peerj-109283-consent_form.pdf, I don’t find partially deny sentences to public the dataset.

Experimental design

• Originality and Scope: n.p.

• Research Questions and Hypotheses: n.p.

• Methodology: n.p.

• Ethics: Ethical approval is explicitly mentioned, and participant confidentiality appears well-protected. I confirmed the IRB of Jinan University, Approval Number: JNUKY-2023-0154.

Validity of the findings

・Data and Analysis: n.p.
・Interpretation: n.p.
・Limitations: n.p.

Additional comments

・What does this sentence mean? L.147. according to the World Health Organization (WHO)(World Health Organization, n.d.).

・Result should not include the interpretations.
ex)
L.258
indicating that obese college students with higher levels of grit tend to exhibit greater physical literacy.
L.260
suggesting that enhanced physical literacy contributes to higher well-being.
L.298-302
These findings demonstrate that while all three subdimensions of physical literacy partially mediate the relationship between grit and well-being, interaction with the environment has the strongest mediating effect, suggesting that enhancing students engagement with their environment may be particularly effective in leveraging grit to improve well-being among obese college students.

<General comments>
I confirmed this study fits well within the aims and scope of the journal. The focus on the mediating role of physical literacy in the relationship between grit and well-being among obese college students addresses a novel and meaningful topic.
This study also confirmed the assumptions of linearity, normality, and sufficient sample size for mediation analysis. Pearson correlation coefficients support linearity, skewness and kurtosis indicate near-normal distributions, and G*Power calculations ensure adequate statistical power. The use of the PROCESS macro (Model 4) with bootstrapping (5,000 resamples) provided robust and statistically significant results (indirect effect = 0.20, 95% CI [0.09, 0.31]).

Therefore, The manuscript is suitable for publication after revisions to ensure raw data.

---

## Round 0.2 · accepted · Accept

Dear Authors,

Congratulations. I appreciate the care you’ve taken in addressing the reviewers’ comments—the manuscript has been significantly improved.

Before finalizing publication, we kindly ask you to review the text and correct a few minor typographical and formatting issues.

Please submit the updated version at your earliest convenience. If you have any questions, feel free to reach out.

We look forward to finalizing your work for publication.

Reviewer 1 ·

Basic reporting

After revision the article reads well.

Experimental design

The design is now clear based upon revisions.

Validity of the findings

No concerns, the over-reaching statements has been dramatically reduced.

Additional comments

I thank the authors for adapting the manuscript to the criticisms and commentary provided.

·

Basic reporting

It is better to spell out words that are not accompanied by an explanation of their abbreviations, such as "M."

Experimental design

I confirm raw data.

Validity of the findings

I confirm these modifications.